# Technical efficiency of primary health care facilities in providing adolescent mental, sexual and reproductive health services in Ghana: A case study of selected districts in the Greater Accra Region

Jacob Novignon[1], Ama Pokuaa Fenny[2]*, Michel Adurayi Amenah[3,4ᵒ], Selasie Addom[3], Annick Gladzah[3,5], Nassirou Ibrahim[6,7], Ludovic Deo Gracias Tapsoba[7,8], Irene A. Agyepong[3,5], Roxanne Borges da Silva[7], Tim Ensor[9]

1 Centre for Social Policy Studies, University of Ghana, Legon, Ghana, 2 Institute of Statistical Social and Economic Research, University of Ghana, Legon, Ghana, 3 Faculty of Public Health, Ghana College of Physicians and Surgeons, Accra, Ghana, 4 Division of Health Research, Lancaster University, Lancaster, United Kingdom, 5 Dodowa Heath Research Centre, Ghana Health Service, Dodowa, Ghana, 6 Laboratory for Studies and Research on Social Dynamics and Local Development, Niamey, Niger, 7 School of Public Health, University of Montreal, Montreal, Canada, 8 Laboratory of Public Health, University of Joseph KI Zerbo,Ouagadougou, Burkina Faso  9 School of Medicine, University of Leeds, Leeds, United Kingdom

ᵒ These authors contributed equally to this work.
* apfenny@ug.edu.gh

## Abstract

Primary healthcare (PHC) facilities have become essential in promoting adolescent healthcare, yet they face resource limitations that hinder their effectiveness. Ensuring the efficient use of available resources has therefore become pertinent. This study assessed the technical efficiency of primary health care facilities in providing adolescent mental sexual and reproductive health (AMSRH) services. Data was collected from 53 PHC facilities drawn from rural and urban locations in four districts in the Greater Accra region using a multi-stage sampling design. Stochastic Frontier Analysis (SFA) was employed to estimate the technical efficiency of each facility in optimizing outputs given available inputs. The findings revealed significant variation in efficiency, ranging from 0.91 to 0.04 with an average score of 0.60. Rural facilities and government-owned health facilities were more efficient compared to their urban and private counterparts. Facilities offering a wider scope of services to adolescents were also more efficient. However, the provision of adolescent mental health services was limited. Efforts should improve efficiency in the use of AMSRH services by properly aligning resource allocation to needs while expanding the range of services available to adolescents.

## 1. Introduction

Adolescence marks a transition phase of human development from childhood to adulthood. This phase of life has been shown to be associated with physical, cognitive and psychosocial

**Data availability statement:** All relevant data are within the manuscript and its Supporting Information files.

**Funding:** This research with grant number [MR/T040203/1] is jointly funded by the UK Medical Research Council (MRC) and the Foreign Commonwealth and Development Office (FCDO) under the MRC/FCDO Concordat agreement, together with the Department of Health and Social Care (DHSC). The funders had no role in study design, data collection and analysis, decision to publish, or preparation of the manuscript.

**Competing interests:** The authors have declared that no competing interests exist.

transformations that may determine the ultimate health state of an individual in adulthood. In this regard, adolescent mental, sexual, and reproductive health (AMSRH) services play a crucial role in addressing the global burden of disease in this population and improving general wellbeing, with mental health conditions like suicide being a leading cause of death among adolescents worldwide, including in sub-Saharan Africa [1,2]. Globally, while adolescent mortality has decreased, morbidity has increased due to persistent issues like injuries, mental health disorders, and chronic diseases [2]. For instance, mental health conditions like suicide have been reported to be among the leading causes of death among adolescents worldwide, including sub-Saharan Africa [1,2].

In Ghana, the recent population and housing census reports that about 20% of the population are adolescents between the ages of 10 and 19 [3]. Despite their substantial population, access to health care remains challenging, posing threats to their health and wellbeing. For example, only 18% of sexually active adolescents aged 15–19 use modern contraceptive methods, and 31% received information about STIs Furthermore, about 14% of adolescent girls aged 15–19 had begun childbearing, with more than half of these pregnancies being unintended [4]. Moreover, while current policies emphasize sexual and reproductive health (SRH), there is a notable lack of focus on adolescent mental health [5]. This gap persists despite evidence linking SRH and mental health, which underscores the need for integrated care.

The poor access to AMSRH services has been attributed to several factors including the lack of diagnostic tools tailored for adolescents, financial constraints (particularly for mental health services), and pervasive stigma surrounding AMSRH [6,7]. Primary healthcare facilities have been identified as a potential enabler of improved healthcare access as they reach a wider population with relatively less resources [9]. Ghana's PHC strategy emphasizes the importance of universal access to essential healthcare services including AMSRH [8]. The strategy aims to provide accessible, affordable, and quality healthcare to all Ghanaians, especially those living in rural and underserved communities.

PHC in Ghana includes health facilities at different levels starting from Community-based Health Planning and Services (CHPS) which are the lowest level of care, health centers, and accredited private clinics. According to the Ghana Health Service (GHS), there are approximately 4,800 health facilities in Ghana, including hospitals, health centers, clinics, and CHPS compounds [4]. Of these, there are about 2,300 CHPS compounds with coverage estimated at 60% of the country's population [4]. By their design, PHC facilities in Ghana have the potential to fill the healthcare delivery gap for adolescents [9,10]. However effective service delivery at this level will require sufficient resource allocation and reforms to address current resource constraint challenges that hinder PHC service delivery [11,12]. While additional resources can be committed, ensuring that the available resources are used efficiently has been showed to improve resource availability [13,14]. Studies have shown that improving efficiency has potential to free up additional resources within the existing resource envelope while improving health outcomes [14,15]. Yet, there is currently no evidence that assesses how technically efficient these primary health facilities use available resources for AMSRH services. Most of the studies on health system efficiency focus on health facilities with a mix of services. Very few studies have focus on technical efficiency in the delivery of specific health services such as malaria [13,14,16–18].

Against this backdrop, the objective of this paper is twofold; (i) to estimate the technical efficiency of primary health facilities in providing adolescent mental, sexual and reproductive health services and (ii) to identify environmental factors that explain the level of technical efficiency. The rest of the paper is organized into four sections with sections two and three presenting the methods and results, respectively, while the last two sections cover the discussion of results and conclusion.

## 2. Materials and methods

### 2.1 Data

AMSRH services are provided by both public and private healthcare facilities and therefore the study sampled both categories of providers. The services included are adolescent sexual and reproductive health services, such as contraception, pregnancy testing and counselling, maternal and child health services, STI testing and treatment, reproductive health education, and safe abortion services, as well as adolescent mental health services.

Primary data was collected from PHC facilities in four districts in the Greater Accra Region of Ghana. This data included information on inputs and outputs related to both ASRH and AMH services provided by the facilities, such as human resources, equipment, and service delivery statistics. The facilities were contacted through the District Health Directorate in each district, which provided a list of all functional primary healthcare facilities. The directorate also supplied the addresses and contact information of the heads of each health facility. Informed written consent was obtained from the heads of the healthcare facilities or their designated representatives before data collection commenced. All participants received comprehensive information about the study's objectives and procedures.

### 2.2 Sampling

We used a multi-stage sampling design to determine the sample size for this study. First, the Greater Accra region was purposefully selected as it contains a combination of urban and rural districts, as well as a diverse population that is representative of the country. Four districts were then purposively selected from the Greater Accra region. Specifically, two rural districts (Ningo Prampram and Shai Osudoku) with rapid urbanization in the periphery bordering the Accra and Tema metropolitan districts and two urban districts (Ga East and La Kwantemang) were included in the sample to allow for differences in service provision between rural and urban areas. In the third and final phase, all functional PHC facilities in the four selected districts were included in the study. Data was collected from a total of 78 facilities (including school-based sick bays) across all four districts. A sub-sample of 53 PHC facilities were included in the efficiency analysis after removing sick bays that did not meet the classification of PHC facilities in Ghana. Of these facilities, 28 (53%) were in rural areas while 25 (47%) were in urban areas.

### 2.3 The stochastic frontier analysis (SFA) model

To measure technical efficiency, we used stochastic frontier analysis (SFA) which was simultaneously proposed by Aigner et al. [19] and Meeusen et al. [20]. The SFA is a parametric approach to assessing efficiency as opposed to nonparametric approaches like data envelopment analysis (DEA) [21]. The analysis begins with the estimation of a production function specified as follows.

$$y_i = x_i^{'}\beta + v_i - u_i \tag{1}$$

Where $y_i$ represents the output of the i[th] decision-making unit (DMUs); $x_i$ is a vector of inputs; $v_i$ is the random disturbance term that accounts for random variations in the production functions of the DMUs. $u_i$ is a non-negative random disturbance term that measures the inefficiency of the DMUs. Both $v_i$ and $u_i$ are assumed to be independent and identically distributed across observations. $u_i$ measures the shortfall of output from its potential maximum frontier given by

$$lny_i^* = x_i^{'}\beta + v_i. \tag{2}$$

Thus;

$$u_i = \left( x_i^{'}\beta + v_i \right) - (x_i^{'}\beta + v_i - u_i) = (lny_i^{*} - lny_i) \tag{3}$$

Therefore, $u_i \times 100\%$ shows the percentage of output lost due to technical inefficiency where a value closer to 100% indicates high inefficiency (this is termed as technical inefficiency index).

Therefore, the expression to obtain the technical efficiency (TE) index of a DMU is arrived at by rearranging equation ([3]) as follows:

$$TE_i = \exp(-u_i) = \frac{\left( x_i^{'}\beta + v_i - u_i \right)}{\left( x_i^{'}\beta + v_i \right)} \geq 0 \tag{4}$$

Equation ([4]) defines technical efficiency as the ratio of the observed output of the $i^{th}$ DMU to the potential maximum output produced by a completely efficient DMU using the same input mix [21]. The value of technical efficiency is between 0 and 1. This implies that the observed output always lies on or below the efficient frontier. A technical efficiency index closer to 1 indicates high technical efficiency while a value closer to zero means less technical efficiency.

The production function is an important component of the SFA as it determines the accuracy of the frontier from which technical efficiency estimates will be computed. While several forms of the production function exist, the Cobb-Douglas and Translog functions are popular in the efficiency literature [22,23]. In this paper, we used the translog function due to its flexibility as it, among others, allows for cross-product terms between inputs hence capturing complementary and substitution effects. The function is specified in this study as follows.

$$\begin{aligned} lny_i = a + \ \alpha lnk_i + \beta lnl_i + \ \delta lnq_i + \alpha\beta lnk_i lnl_i + \alpha\delta lnk_i lnq_i \\ + \ \beta\delta lnl_i lnq_i \ + \alpha\alpha \left( lnk_i \right)^2 + \ \beta\beta \left( lnl_i \right)^2 \ + \ \delta\delta \left( lnq_i \right)^2 + \ \varepsilon_i \end{aligned} \tag{5}$$

Where $y_i$ is the output of the i-th DMU (health facilities), $k$ represents capital of the DMUs (number of beds in the facility), $l$ represents labour inputs of the DMUs (time spent by clinical staff, time spent by nonclinical staff), and $q$ represents other potential determinants of output of the facilities (such as facility type, gender of head of the facility among others) whiles α, β and δ are estimable parameter. ε is the composite error term made of $v$ (the random disturbance term) and $u$ (the inefficiency term). The production function reports Lamda (λ) which is the ratio of the standard deviation of the inefficiency component ($u$) to the standard deviation of the idiosyncratic component $(v)$, and represents the technical efficiency of the production process. Lamda greater than 1 means the standard deviation of the inefficiency component is higher than the standard deviation of the random disturbance term, suggesting the presence of inefficiency in the production process.

Technical efficiency can be computed from the output or input orientations. The output orientation holds inputs constant and determines how outputs could potentially be expanded while input orientation keeps outputs fixed and explores possible reduction in inputs [24–27]. In this study, we used the output orientation given that the health facilities only receive a set of resources and the decision under their control is usually how much outputs can be produced with the available resources. Moreover, the ultimate objective of health facilities is to expand output rather than limiting available resources.

In the second stage analysis, we estimate how health facility and environmental factors are associated with the levels of technical efficiency estimated in the first stage. We used the Tobit regression technique which belongs to the family of limited variable models. The Tobit model was appropriate as it allows for continuous dependent variables that are observable only

within an interval. In our case, the technical efficiency scores are limited within a 0–1 interval [28]. The estimated equation is presented in equation (6).

$$te_i = \alpha_0 + \alpha_1 hf_i + \alpha_2 comm_i + \varepsilon_i \qquad (6)$$

Where *te* is the estimated technical efficiency scores, *hf* is health facility characteristics and *comm* is community factors.

## 2.6 Variable description

Table 1 presents a description of output, input, and other environmental variables included in the analysis.

The variables included in the model were selected based on their relevance to ASRH services and their use in analogous studies on healthcare facility efficiency. Key inputs like clinical and non-clinical staff time, beds, lab tests, and medicines reflect resource allocation, while outputs such as outpatient attendance measure service utilization [26,29]. Environmental variables like facility location, type (public vs. private), and leadership gender help account for external factors influencing efficiency. Additionally, economic factors such as the Gini coefficient and health insurance coverage provide context for disparities in healthcare access, drawing from similar studies that analyse health system performance.

## 2.7 Estimation issues

An important concern with efficiency analysis that focuses on a sub-set of services produced by a given organization is the attribution of inputs to these services. In the case of SRH services in Ghana, health workers in PHC facilities are also involved in providing services across different disease conditions. In this regard, for the key inputs, we either adjust them with the average time spent per patient in providing AMSRH services or we use the variables as a proportion of AMSRH services delivered at the facility. The output variable we used is also subject to bias relating to the mix of services used by an adolescent reporting for OPD and the severity of service. For example, while two facilities may have the same number of SRH OPD patients each year, the intensity or quantity of the services they seek may be different, and therefore resource needs will not be the same. Some authors have recommended that outputs used in efficiency analysis of this nature be adjusted to reflect these nuances [30]. In practice, this is a difficult undertaking as data is typically limited. In this paper, we used the average time spent on adolescents in the facility to adjust the outcome variable. We consider this a reasonable way to account for service intensity (Service intensity refers to the complexity and resource demands of a health service. It accounts for variations in time, personnel, and equipment required, ensuring fair comparisons of facility efficiency). We, however, do acknowledge that our approach does not completely purge the variable of this potential bias.

## 2.8 Ethics

Ethical approvals for this study were obtained from the Ghana Health Service Ethics Review Committee (GHS ERC 021/05/21) and the University of Leeds Medical Research Ethics Committee (MREC 21–010 External - AdoWA project). Written informed consent was obtained from the heads of the healthcare facilities or their designated representatives prior to data collection. All participants were provided with detailed information regarding the study's objectives and procedures. The interviews were conducted to gather institutional-level data, and no personal data from patients were collected. Data were anonymized to ensure confidentiality, and all procedures adhered to the ethical standards set forth by the approving committees

**Table 1. Summary of input and output variables.**

| Variable | Measurement | Definition |
|---|---|---|
| **Outputs** | | |
| **Outpatient attendance** | Total number of adolescent outpatients receiving Adolescent Sexual and Reproductive Health (ASRH) services multiplied by the average time spent by clinical staff on an adolescent. | Number of adolescent patients receiving ASRH services, adjusted by the average time spent by clinical staff to account for the severity of outpatient care in the facility. |
| **Inputs** | | |
| **Clinical staff time** | Total number of clinical staff in the health facility multiplied by the average clinical time (in minutes) spent on an adolescent per week | This variable is defined as the amount of time committed by clinical staff to providing ASRH services in a health facility. It also serves a key input in the production of adolescent healthcare services. |
| **Non-clinical staff time** | Total number of non-clinical staff in the health facility multiplied by the average time (in minutes) spent per week on adolescents by non-clinical staff. | The variable captures the amount of time devoted to adolescent healthcare services by non-clinical staff |
| **Beds** | Total number of beds in the facility. | Total number of beds in the facility adjusted by the proportion of adolescent visits. |
| **Laboratory tests** | Total number of laboratory tests administered to adolescents in health facilities multiplied by the proportion of OPD cases that are related to ASRH. | Total number of laboratory tests performed for adolescent patients seeking ASRH-specific services. |
| **Medicines** | Total number of medicine packs disbursed to adolescents in health facilities multiplied by the proportion of OPD cases that are ASRH. | Total number of medicine packs dispensed to adolescents seeking ASRH services. |
| **Environmental factors** | | |
| **Computer availability** | Dummy variable: 1 = facility has a functional computer, 0 = otherwise. | Indicates whether the facility has a functional computer. |
| **Time spent by nurses on adolescents** | Average time spent by a nurse on an adolescent per visit to the facility. | Represents the average time a nurse spends with an adolescent patient per visit. |
| **Ratio of clinical staff to non-clinical staff** | Number of clinical staff in the health facility divided by the number of non-clinical staff in 2021. | Represents the ratio of clinical staff to non-clinical staff in the facility. |
| **Ratio of clinical staff to total staff** | Number of clinical staff in the health facility divided by the total staff in the health facility in 2021. | Represents the ratio of clinical staff to the total number of staff in the facility. |
| **Facility has a laboratory** | Dummy variable: 1 = facility has a functional laboratory, 0 = otherwise. | Indicates whether the health facility has a functional laboratory. |
| **Facility has a pharmacy** | Dummy variable: 1 = facility has a functional pharmacy, 0 = otherwise. | Indicates whether the health facility has a functional pharmacy. |
| **Facility has running vehicles** | Dummy variable: 1 = facility has a running vehicle, 0 = otherwise. | Indicates whether the health facility has a running vehicle. |
| **Availability of consultation rooms** | Total number of consultation rooms in the facility. | Total number of consultation rooms available in the facility. |
| **Access to electricity** | Dummy variable: 1 = facility has electricity, 0 = no electricity. | Indicates whether the facility has an electricity supply. |
| **Availability of water supply** | Dummy variable: 1 = facility has regular water supply, 0 = no regular water supply. | Indicates whether the facility has a regular water supply. |
| **Insurance coverage** | Proportion of the district population with health insurance coverage. | Represents the proportion of the district population that is covered by health insurance. |
| **Percentage of adolescent population** | Percentage of the population of the catchment area of the health facility who are adolescents. | Indicates the percentage of the adolescent population within the facility's catchment area. |
| **Location of facility (Urban/Rural)** | Dummy variable: 1 = urban facility, 0 = rural facility. | Indicates whether the facility is located in an urban or rural area. |
| **Facility type (public/private)** | Dummy variable: 1 = public facility, 0 = private facility. | Indicates whether the facility is public or private. |
| **Gender of facility head** | Dummy variable: 1 = male head, 0 = female head. | Indicates the gender of the facility head. |
| **Gini coefficient** | District-level Gini coefficient for facilities located in that district. | Measures economic inequality within the district where the facility is located. |
| **Literacy rate** | Average literacy rate in the district allocated to all facilities within the district. | Represents the average literacy rate in the district, applied to all facilities within that district. |

## 3. Results

Descriptive statistics of key variables included in the analysis are presented in Table 2. The results show that approximately 87% of the health facilities had laboratories, 66% of them had a functional computer and 53% were in rural areas. While 55% of the DMUs were government facilities, 72% had female heads. About 64% were health centers and CHPS compounds while the other 36% were primary-level clinics and hospitals. The table reports that facilities had 8 beds on average. On average, about 26 percent of the population in a catchment area are adolescents. About 94 percent of the facilities have a pharmacy, 30 percent of them have running vehicles while 79 percent of them have regular supply of water.

Each facility provided approximately 243 ASRH services on average in 2021. Also, there are approximately 2 non-clinical staff per facility. Clinical staff spend about 4 hours and 15 minutes on adolescents per week and each clinical staff provides about 23 ASRH services per year while non-clinical staff spends about 7 minutes per adolescents per week.

While the study set out to assess both AMH and sexual and ASRH services, the data reveal a stark gap in the provision of AMH services. On average, the number of OPD services provided for adolescent mental health was just 0.71 per facility, compared to 243 ASRH services. This finding highlights the limited availability of mental health services, with few primary care facilities having mental health professionals specifically oriented to provide adolescent mental health services. The limited provision of AMH services is consistent with previous findings

**Table 2. Descriptive statistics.**

| Variable | Mean | Std. Dev. | Minimum | Maximum |
|---|---|---|---|---|
| Clinical staff time (minutes per week) | 248.94 | 577.94 | 5.83 | 3819.14 |
| Non-clinical staff time (minutes per week) | 7.07 | 13.61 | .91 | 92.65 |
| Total beds | 8 | 6.66 | 1 | 39 |
| Medicines (number of packs) | 201.90 | 612.84 | .15 | 3686.64 |
| Laboratory tests (number of tests) | 73.99 | 303.48 | .012 | 1768.23 |
| OPD service for ASRH (number of OPD attendance) | 242.98 | 637.17 | 4 | 4381 |
| Number of OPD services for AMH | 0.71 | 1.25 | 0 | 12 |
| Number of Services provided per clinical staff | 22.64 | 23.60 | .44 | 110 |
| Location of facility (urban) | .47 | .50 | 0 | 1 |
| Facility type (public) | .55 | .50 | 0 | 1 |
| Gender of facility head (Male) | .28 | .46 | 0 | 1 |
| Gini coefficient | 37.31 | 1.97 | 35.5 | 40.1 |
| Health center/CHPS | .64 | .48 | 0 | 1 |
| Facility has a computer | .66 | .48 | 0 | 1 |
| Literacy rate | .82 | .08 | .75 | .92 |
| Ratio of clinical staff to non-clinical staff | .57 | .44 | .06 | 2.02 |
| Facility has a laboratory | .87 | .34 | 0 | 1 |
| Number of consultation room | 1.64 | 1.09 | 1 | 7 |
| Services provide per non-clinical staff | 55.7 | 71.09 | 1.25 | 335.4 |
| Health insurance coverage | .63 | .07 | .55 | .71 |
| Percentage of adolescent population | 25.84 | 8.26 | 10 | 67 |
| Does facility have electricity supply | .98 | .14 | 0 | 1 |
| Does facility have a Pharmacy | .94 | .23 | 0 | 1 |
| Does the facility have running vehicles (Yes=1) | .30 | .46 | 0 | 1 |
| Is there regular supply of water at this facility (Yes =1) | .79 | .41 | 0 | 1 |

that mental health care in these facilities is marginalized, and even when available, is not specialized for adolescent needs.

Table 3 shows the estimated production frontier for the DMUs. The estimated parameters are acceptable for the model. Lambda is reasonably high and significant in the model indicating the presence of inefficiency in the estimated production function (thus inefficiency among the DMUs in the study). The variance, which is decomposed into sigma_u (inefficiency term) and sigma_v (random error term), shows that the inefficiency term dominates the random error term with a Lamda ($\lambda$) value greater than 1 and statistically significant.

**Table 3. SFA translog production function regression results.**

|  | Coefficients |
|---|---|
| Clinical staff time per adolescent | 1.009** |
|  | (0.512) |
| Non-clinical staff time per adolescent | −0.273 |
|  | (0.788) |
| Total beds | 1.975** |
|  | (0.856) |
| AMSRH medicines disbursed to adolescent | 0.282 |
|  | (0.297) |
| AMSRH lab tests conducted for adolescent | 0.682 |
|  | (0.494) |
| Clinical staff time X non-clinical staff time | 0.280 |
|  | (0.181) |
| Clinical staff time X total beds | 0.173 |
|  | (0.220) |
| Clinical staff time X AMSRH medicines disbursed | 0.017 |
|  | (0.053) |
| Clinical staff time X AMSRH lab tests conducted | 0.031 |
|  | (0.063) |
| Non-clinical staff time X by total beds | −0.385** |
|  | (0.173) |
| Non-clinical staff time X AMSRH medicines disbursed | −0.086 |
|  | (0.081) |
| Non-clinical staff time X AMSRH lab tests | 0.159* |
|  | (0.088) |
| Total beds X AMSRH medicines disbursed | 0.001 |
|  | (0.101) |
| Total beds X AMSRH lab tests | −0.197 |
|  | (0.139) |
| AMSRH medicines disbursed X total lab tests | −0.233*** |
|  | (0.087) |
| Clinical staff time per adolescent square | −0.107 |
|  | (0.073) |
| Non-clinical staff time per adolescent square | −0.219* |
|  | (0.133) |
| Total beds square | −0.456** |
|  | (0.226) |
| AMSRH medicines disbursed to adolescent square | 0.078 |
|  | (0.048) |

**Table 3.** (Continued)

| | Coefficients |
|---|---|
| AMSRH lab tests conducted for adolescent square | 0.112*** |
| | (0.037) |
| Constant | 0.090 |
| | (1.661) |
| Sigma_U | 0. 693*** |
| | (0.202) |
| Sigma_V | 0.405*** |
| | (0.145) |
| Lamda | 1.711*** |
| | (0.323) |
| N | 53 |

Note: *p<0.10, **p<0.05, ***p<0.01. Standard errors are reported in parenthesis.

Table 4 presents the mean efficiency scores of the facilities across facility characteristics. Overall, average technical efficiency was estimated to be about 0.6 suggesting that on average, about 40% of health facility resources could have been saved while producing the observed levels of output. The disaggregated results show that primary healthcare facilities in rural (0.63) areas are more efficient than their counterparts in urban (0.56). Government health facilities are more efficient (0.63) than private healthcare facilities (0.56). There is relatively marginal difference in efficiency between female (0.60) and male (0.59) -headed facilities. Health centers and CHPS compounds are more efficient (0.63) than clinics (0.55). Facilities in the Shai Osudoku and Ningo Prampram districts lead in terms of efficiency (0.62) followed by Ga East district (0.56) and La Kwantanang (0.55). Moreover, facilities that have a laboratory (0.60) are slightly more efficient than facilities that do not have a laboratory (0.59). The reported differences across facility characteristics were all statistically significant as indicated by the 95% confidence interval.

Table 5 presents regression results on the environmental determinants of technical efficiency. A positive association suggests the variable is an enabler of efficiency while a negative association suggests reduction in efficiency. The regression results show that the number of ASRH services provided per clinical staff, number of consultation rooms in a facility and proportion of the district population with health insurance all have a positive and statistically significant association with technical efficiency in using ASRH resources. While the log of Gini index and literacy rate at the district level have significant negative effects on technical efficiency. The other determinants of efficiency in the model are not statistically significant at the 5% conventional level.

## 3.4 Discussion

The study aimed to assess the technical efficiency of primary health facilities in providing AMH and ASRH services. The findings show a notable lack of AMH services, with primary care services largely focused on ASRH. The average technical efficiency score of facilities was 0.60, meaning that with current input levels, output can be expanded by an average of 40%. The average efficiency score was higher among rural facilities compared to their urban counterparts. Government facilities were also more efficient relative to private facilities. These findings echo prior research that confirm the presence of inefficiency across health facilities in general health service delivery [13,14,18] and specific disease conditions

**Table 4. Mean of efficiency scores by facility characteristics.**

|  | Mean | Standard error | 95% confidence interval | |
|---|---|---|---|---|
| Location of facility |  |  |  |  |
| Rural | 0.630 | 0.040 | 0.550 | 0.709 |
| Urban | 0.561 | 0.045 | 0.471 | 0.651 |
| Type of facility |  |  |  |  |
| Private | 0.561 | 0.048 | 0.465 | 0.657 |
| Government | 0.627 | 0.037 | 0.553 | 0.702 |
| Facility has a computer |  |  |  |  |
| No | 0.631 | 0.050 | 0.532 | 0.731 |
| Yes | 0.580 | 0.037 | 0.505 | 0.655 |
| Facility has a laboratory |  |  |  |  |
| No | 0.593 | 0.105 | 0.383 | 0.804 |
| Yes | 0.598 | 0.031 | 0.536 | 0.660 |
| Clinic or Health center |  |  |  |  |
| Clinic/hospitals | 0.548 | 0.057 | 0.434 | 0.662 |
| Health center/CHPS | 0.625 | 0.034 | 0.557 | 0.692 |
| Gender of facility head |  |  |  |  |
| Female | 0.602 | 0.036 | 0.530 | 0.675 |
| Male | 0.585 | 0.054 | 0.477 | 0.693 |
| District |  |  |  |  |
| Ga East | 0.560 | 0.080 | 0.399 | 0.721 |
| Shia Osudoku | 0.622 | 0.058 | 0.506 | 0.738 |
| Ningo Prampram | 0.615 | 0.045 | 0.525 | 0.704 |
| La Kwantanang | 0.553 | 0.074 | 0.404 | 0.702 |
| Total average | 0.593 | 0.052 | 0.489 | 0.697 |

like malaria [16]. Our results also confirm previous findings by Novignon & Nonvignon [31] suggesting that countries like Ghana could greatly enhance primary healthcare delivery and utilization by increasing technical efficiency. These findings have broader implications for resource efficiency in low-resource settings. The presence of inefficiency suggests that there is significant potential to optimize the use of existing resources. By improving efficiency, cost savings can be achieved, enabling additional outputs to be generated within the same resource envelope. This is particularly important in resource-constrained environments, where optimizing and reallocating limited inputs can enhance healthcare delivery.

The results imply that there is the possibility of producing more outputs within the existing resource envelope. This may be achieved by reviewing the input mix at PHC facilities to ensure that the productivity of these inputs is improved by deploying them where they are most needed. The gains from improved efficiency could result in generating additional resources within existing envelope that could be repurposed within the health sector [15]. Observations from the fieldwork also underscore the estimated level of inefficiencies. We heard from health workers that due to the sensitive nature of sexual and reproductive health services provided at the facilities, adolescents find it difficult to seek care at these facilities. Consequently, some adolescents prefer to self-medicate or avoid seeking care altogether. This means existing resources may be under-utilized as fewer adolescents seek services from formal facilities.

**Table 5. Determinants of technical efficiency.**

| Variables | Coefficients |
|---|---|
| Services provided per clinical staff | 0.005* |
|  | (0.002) |
| Location of facility (Urban=1) | −0.112 |
|  | (0.082) |
| Facility type (Government=1) | −0.137 |
|  | (0.111) |
| Gender of head of facility (Male=1) | 0.051 |
|  | (0.058) |
| Log of Gini index | −43.340* |
|  | (19.305) |
| Facility is a health center | 0.191 |
|  | (0.096) |
| Facility has a computer | 0.024 |
|  | (0.063) |
| Literacy rate | −21.119* |
|  | (9.423) |
| Ratio of clinical staff to non-clinical staff | −0.040 |
|  | (0.077) |
| Number of consulting rooms in your facility | 0.074* |
|  | (0.026) |
| Services provided per non-clinical staff | 0.000 |
|  | (0.001) |
| Insurance | 18.852* |
|  | (8.582) |
| Percentage of the population that are adolescent | −0.000 |
|  | (0.004) |
| Does facility have electricity supply | 0.165 |
|  | (0.239) |
| Does the facility have a functional pharmacy | −0.271 |
|  | (0.141) |
| Does the facility have running vehicles | 0.027 |
|  | (0.076) |
| Is there regular supply of water at this facility? | −0.114 |
|  | (0.067) |
| Constant | 162.906* |
|  | (72.250) |
| N | 53.000 |
| r2_p | −2.611 |

Note: *indicate statistical significance at 5%. Standard errors in parenthesis.

In the second stage where we evaluate the potential association between facility characteristics, external environmental factors and inefficiency, we find some interesting results that could guide policy reforms to improve technical efficiency. For instance, the findings suggest that providing a wider range of ASRH services is associated with higher technical efficiency. This is justifiable as facilities that have more services can provide more comprehensive services and save resources. Resource use may not be optimal in cases where patients seek care

for different services across different facilities. The findings also confirm that having more consulting rooms in the facility reduced inefficiency. Indeed, the availability of relatively more consulting rooms has the potential to reduce waiting time and, ultimately, improve efficiency. While PHC facilities are mostly designed as small structures with few consulting rooms, dedicated spaces could be created to meet the peculiar needs of adolescents and improve efficiency in resource use.

With regards to environmental factors external to the health facility, the results show that better economic environment (measured with the level of economic inequality) and reduced financial barriers to health care (measured with health insurance coverage) are associated with improved technical efficiency. Several studies have shown how poverty and inequality limit health care seeking and the role of insurance in enhancing health care utilization [32–36]. The findings support government efforts to improve livelihood and reduce financial barriers to health care through programmes like the Livelihood Empowerment Against Poverty (LEAP) and the National health Insurance Scheme (NHIS). Scaling up these interventions to ensure they are effective and reach the poor and vulnerable adolescents will encourage health seeking at formal facilities, thereby reducing resource redundancy at health facilities.

There are important strengths and limitations of the study worth discussing. The key strength of this study lies in its attempt to apply technical efficiency techniques to ASRH services. As mentioned earlier, technical efficiency is typically applied to health facilities as a whole and does not single out specific services. This is mostly due to the lack of data or difficulty in separating inputs and outputs for specific services within health facilities that provide a mix of services. To the best of our knowledge, this is the first effort to estimate the technical efficiency of PHC facilities in providing ASRH services in Ghana. Despite its strength, the study was limited in scope as it was confined to four districts in the Greater Accra Region, which may limit the generalizability of the findings to other regions in Ghana or other countries. Also, while the study included different types of health facilities, other influential factors such as staff qualifications, workload, or patient satisfaction were not accounted for in the analysis. Finally, our inability to include mental health services in our measure of efficiency is a limitation of the study. Future studies on technical efficiency of AMSRH resource use should consider addressing these limitations.

## 4  Conclusion

This study provides critical insights into the efficiency of primary healthcare (PHC) facilities in delivering adolescent sexual and reproductive health (ASRH) services in the Greater Accra Region of Ghana. Our findings show significant inefficiencies, with about 40% of resources being underutilized in ASRH service delivery. In contrast, adolescent mental health services are minimally provided, highlighting a significant gap in comprehensive adolescent care.

The results suggest that facilities with a broader scope of ASRH services and more consulting rooms are more efficient. External factors, such as reducing financial barriers through health insurance coverage and addressing economic inequality, are also associated with better technical efficiency in health facilities.

These findings highlight the potential to improve adolescent health service delivery by optimizing resource allocation and addressing both internal facility characteristics and external environmental factors. Future interventions should focus on creating more adolescent-friendly spaces, integrating mental health services, and expanding financial access to healthcare to improve overall efficiency in primary health facilities.

# Appendix

**Table A1. Summary of Input, Output, and Environmental Variables.**

| Category | Variable | Definition |
|---|---|---|
| **Inputs** | **Clinical staff time** | Total time spent by clinical staff (doctors, nurses) attending to adolescents, calculated as staff count multiplied by average time spent per adolescent. |
| | **Non-clinical staff time** | Total time spent by non-clinical staff (administrative/support staff) attending to adolescent-related services. |
| | **Beds** | Total number of beds available for adolescent patients in the facility. |
| | **Laboratory tests** | Total number of laboratory tests performed for adolescent patients, adjusted for AMSRH-specific services. |
| | **Medicines** | Total number of medications dispensed to adolescents, adjusted for AMSRH-specific services. |
| **Outputs** | **Outpatient attendance** | Number of adolescent patients receiving AMSRH services, adjusted by the average time spent by clinical staff. |
| **Environmental Factors** | **Location of facility (Urban/Rural)** | Dummy variable: 1 = urban facility, 0 = rural facility. |
| | **Facility type (Public/Private)** | Dummy variable: 1 = public facility, 0 = private facility. |
| | **Gender of facility head** | Dummy variable: 1 = male head, 0 = female head. |
| | **Gini coefficient** | Measure of economic inequality within the district where the facility is located. |
| | **Literacy rate** | Average literacy rate in the district, applied to the facilities within that district. |
| | **Availability of consultation rooms** | Total number of consultation rooms in the facility. |
| | **Access to electricity** | Dummy variable: 1 = facility has electricity, 0 = no electricity. |
| | **Availability of water supply** | Dummy variable: 1 = facility has regular water supply, 0 = no regular water supply. |
| | **Insurance coverage** | Proportion of the district population covered by health insurance. |
| | **Percentage of adolescent population** | Percentage of the facility's catchment area population that is adolescent. |

Note: This appendix summarizes the key input, output, and environmental variables used in the study to assess the technical efficiency of primary healthcare facilities in providing Adolescent Mental, Sexual, and Reproductive Health (AMSRH) services.

# Supporting information

**S1 Data. Adowa.**
(XLS)

# Acknowledgments

Not applicable.

# Author contributions

**Conceptualization:** Ama Pokuaa Fenny, Jacob Novignon, Selassie Adom, Annick Gladzah, Nassirou Ibrahim, Ludovic Deo Gracias Tapsoba, Irene A. Agyepong, Roxanne Borges Da Silva, Tim Ensor.

**Data curation:** Ama Pokuaa Fenny, Jacob Novignon, Michel Adurayi Amenah, Annick Gladzah.

**Formal analysis:** Jacob Novignon, Michel Adurayi Amenah, Annick Gladzah.

**Funding acquisition:** Ama Pokuaa Fenny, Irene A. Agyepong, Roxanne Borges Da Silva, Tim Ensor.

**Investigation:** Ama Pokuaa Fenny, Jacob Novignon, Michel Adurayi Amenah.

**Methodology:** Ama Pokuaa Fenny, Jacob Novignon, Michel Adurayi Amenah, Selassie Adom, Annick Gladzah, Nassirou Ibrahim, Ludovic Deo Gracias Tapsoba, Irene A. Agyepong, Roxanne Borges Da Silva, Tim Ensor.

**Project administration:** Ama Pokuaa Fenny, Irene A. Agyepong.

**Supervision:** Selassie Adom, Annick Gladzah, Irene A. Agyepong, Roxanne Borges Da Silva, Tim Ensor.

**Validation:** Ama Pokuaa Fenny, Jacob Novignon, Michel Adurayi Amenah, Selassie Adom, Annick Gladzah, Nassirou Ibrahim, Ludovic Deo Gracias Tapsoba, Irene A. Agyepong, Roxanne Borges Da Silva, Tim Ensor.

**Visualization:** Ama Pokuaa Fenny, Jacob Novignon, Michel Adurayi Amenah, Selassie Adom, Annick Gladzah, Nassirou Ibrahim, Ludovic Deo Gracias Tapsoba, Tim Ensor.

**Writing – original draft:** Ama Pokuaa Fenny, Jacob Novignon, Michel Adurayi Amenah, Selassie Adom.

**Writing – review & editing:** Ama Pokuaa Fenny, Jacob Novignon, Michel Adurayi Amenah, Selassie Adom, Annick Gladzah, Nassirou Ibrahim, Ludovic Deo Gracias Tapsoba, Irene A. Agyepong, Roxanne Borges Da Silva, Tim Ensor.

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
