## [Decision Letter · Decision Letter 0]

18 Sep 2024

PONE-D-24-09524Technical Efficiency of Primary Health Facilities in Providing Adolescent Mental, Sexual and Reproductive Health Services in Ghana: a case study of selected districts in the Greater Accra RegionPLOS ONE

Dear Dr.  Fenny,

Thank you for submitting your manuscript to PLOS ONE. After careful consideration, we feel that it has merit but does not fully meet PLOS ONE’s publication criteria as it currently stands. Therefore, we invite you to submit a revised version of the manuscript that addresses the points raised during the review process. Please submit your revised manuscript by Nov 02 2024 11:59PM. If you will need more time than this to complete your revisions, please reply to this message or contact the journal office at plosone@plos.org. Please include the following items when submitting your revised manuscript:

We look forward to receiving your revised manuscript.

Kind regards,

Yitagesu Habtu Aweke, Ph.D

Academic Editor

PLOS ONE

Journal Requirements: When submitting your revision, we need you to address these additional requirements. 1. Please ensure that your manuscript meets PLOS ONE's style requirements, including those for file naming. The PLOS ONE style templates can be found at https://journals.plos.org/plosone/s/file?id=wjVg/PLOSOne_formatting_sample_main_body.pdf and https://journals.plos.org/plosone/s/file?id=ba62/PLOSOne_formatting_sample_title_authors_affiliations.pdf 2. We suggest you thoroughly copyedit your manuscript for language usage, spelling, and grammar. If you do not know anyone who can help you do this, you may wish to consider employing a professional scientific editing service.  The American Journal Experts (AJE) (https://www.aje.com/) is one such service that has extensive experience helping authors meet PLOS guidelines and can provide language editing, translation, manuscript formatting, and figure formatting to ensure your manuscript meets our submission guidelines. Please note that having the manuscript copyedited by AJE or any other editing services does not guarantee selection for peer review or acceptance for publication.  Upon resubmission, please provide the following: The name of the colleague or the details of the professional service that edited your manuscript A copy of your manuscript showing your changes by either highlighting them or using track changes (uploaded as a *supporting information* file) A clean copy of the edited manuscript (uploaded as the new *manuscript* file)” 3. Thank you for stating the following financial disclosure: "This research with grant number [MR/T040203/1] is jointly funded by the UK Medical Research Council (MRC) and the Foreign Commonwealth and Development Office (FCDO) under the MRC/FCDO Concordat agreement, together with the Department of Health and Social Care (DHSC). " Please state what role the funders took in the study.  If the funders had no role, please state: ""The funders had no role in study design, data collection and analysis, decision to publish, or preparation of the manuscript."" If this statement is not correct you must amend it as needed. Please include this amended Role of Funder statement in your cover letter; we will change the online submission form on your behalf. 4. We note that your Data Availability Statement is currently as follows: All relevant data are within the manuscript and its Supporting Information files. Please confirm at this time whether or not your submission contains all raw data required to replicate the results of your study. Authors must share the “minimal data set” for their submission. PLOS defines the minimal data set to consist of the data required to replicate all study findings reported in the article, as well as related metadata and methods (https://journals.plos.org/plosone/s/data-availability#loc-minimal-data-set-definition). For example, authors should submit the following data: - The values behind the means, standard deviations and other measures reported;- The values used to build graphs;- The points extracted from images for analysis. Authors do not need to submit their entire data set if only a portion of the data was used in the reported study. If your submission does not contain these data, please either upload them as Supporting Information files or deposit them to a stable, public repository and provide us with the relevant URLs, DOIs, or accession numbers. For a list of recommended repositories, please see https://journals.plos.org/plosone/s/recommended-repositories. If there are ethical or legal restrictions on sharing a de-identified data set, please explain them in detail (e.g., data contain potentially sensitive information, data are owned by a third-party organization, etc.) and who has imposed them (e.g., an ethics committee). Please also provide contact information for a data access committee, ethics committee, or other institutional body to which data requests may be sent. If data are owned by a third party, please indicate how others may request data access. 5. Please include your full ethics statement in the ‘Methods’ section of your manuscript file. In your statement, please include the full name of the IRB or ethics committee who approved or waived your study, as well as whether or not you obtained informed written or verbal consent. If consent was waived for your study, please include this information in your statement as well.

Reviewers' comments:

Reviewer's Responses to Questions

**Comments to the Author**

1. Is the manuscript technically sound, and do the data support the conclusions?

Reviewer #1: Partly

Reviewer #2: Yes

Reviewer #3: Yes

2. Has the statistical analysis been performed appropriately and rigorously? 

Reviewer #1: Yes

Reviewer #2: I Don't Know

Reviewer #3: Yes

3. Have the authors made all data underlying the findings in their manuscript fully available?

Reviewer #1: Yes

Reviewer #2: No

Reviewer #3: No

4. Is the manuscript presented in an intelligible fashion and written in standard English?

Reviewer #1: No

Reviewer #2: Yes

Reviewer #3: Yes

5. Review Comments to the Author

Reviewer #1: Title need to be coincide more and easily understandable. In title it is mentioned Primary Health Facilities, rather it is better to mention Primary Health Care Facilities. Title might be "Efficiency of Primary Health Care Facilities in Providing Adolescent Health Services in Greater Accra, Ghana: a case study"

Abstract section can be more shorter. In abstract the background portion is large enough. It is better to prepare one sentence like " There are gaps in access to and use of AMSRH services, as well as in the quality and efficiency of care provided in many low- and middle-income countries like Ghana." The objective need to be rephrase. To assess is better to mention rather to estimate.

Introduction: This section is also very large. Please avoid to provide similar information in several sentences. Rather provide more references with one sentence. In objective there is mentioned the environmental factor, which can be inserted in title also.

Methodology: How many private facilities and public facilities are selected. How many facilities are selected from urban areas and rural areas need to be mentioned. The process of data collection need to be mentioned more elaborately. Who collect data from whom, and how the consent was taken, who provide permission to assess data need to be added in methodology.

Result: p value need to be added in Table 2. There in huge differences in services providing in different health facilities. So how many facilities providing - the best services, good services, Average services, poor services, and very poor services (Likert scale) can be incorporated. There are some abbreviation used without mentioning their elaboration like OPD, SRH, STIs etc.

Conclusion: This section will more to be reflected from the findings. The conclusion is more generic rather specific. So it could be improved based on the specific findings of the paper.

Reviewer #2: This article has the potential to contribute both strategy and knowledge to future efforts to evaluate and improve health system technical efficiency, especially in low-resource settings. However, as it stands, the work fails to frame the study methods and findings with this potential impact in mind. While I am interested in the “possibility of producing more outputs within the existing resource envelope”, I do not feel that this paper provides a path towards this goal.

In revising this manuscript, I urge the authors to consider the following:

1) Refocus the Introduction to center around resource limitations and the need to maximize AMSRH resources given the demand/burden of disease (draw data from GBD).

2) Define inputs and outputs earlier in the methods and use them to define the analysis in layman’s terms. I am not familiar with SFA so I will defer to the other reviewers on this analysis.

3) Do not discuss the paucity of mental health care in Methods; that is Results.

4) Start the discussion with one paragraph summary of results then describe i) how the methods may be used to evaluate technical efficiency in other low-resource settings and/or other specializations and ii) how the results may be used to improve technical efficiency, in other words, advise where efforts to improve resource efficiency should be focused. Be specific.

5) De-emphasize privacy from the Discussion unless you have additional data on privacy to contribute to the Results.

I thank the authors for their efforts and wish them the best of luck in future submissions.

Reviewer #3: Please see additional file for comments.

This study estimates the technical efficiency of adolescent sexual and reproductive health services among healthcare facilities in Greater Accra Region, Ghana and their environmental determinants. The authors use stochastic frontier analysis to estimate the technical efficiency of each healthcare facility, then multivariable regression analysis to assess the statistical relationships between environmental characteristics and technical efficiency. The authors find that healthcare facilities provide adolescent sexual and reproductive health services at approximately 60% efficiency, but with significant heterogeneity across the sample. Services covered by clinical staff, insurance coverage, the Gini Index, and the literacy rate were all statistically significant. This research provides an assessment of technical efficiency of specific services that are vital for the growing youth population in Ghana, as well as evidence for future policies that could improve the efficiency of those services if implemented.

6. PLOS authors have the option to publish the peer review history of their article (what does this mean? ). If published, this will include your full peer review and any attached files.

**Do you want your identity to be public for this peer review?** For information about this choice, including consent withdrawal, please see our Privacy Policy .

Reviewer #1: No

Reviewer #2: No

Reviewer #3: No

---

## [Author Response · Author response to Decision Letter 0]

30 Oct 2024

The response to reviewers has been attached as a separate document.

---

## [Decision Letter · Decision Letter 1]

17 Jan 2025

PONE-D-24-09524R1Technical Efficiency of Primary Health Facilities in Providing Adolescent Mental, Sexual and Reproductive Health Services in Ghana: a case study of selected districts in the Greater Accra RegionPLOS ONE

Dear Dr. Fenny,

Thank you for submitting your manuscript to PLOS ONE. After careful consideration, we feel that it has merit but does not fully meet PLOS ONE’s publication criteria as it currently stands. Therefore, we invited you to submit the revised version. Please submit your revised manuscript by Mar 03 2025 11:59PM. If you will need more time than this to complete your revisions, please reply to this message or contact the journal office at plosone@plos.org . Please include the following items when submitting your revised manuscript:

We look forward to receiving your revised manuscript.

Kind regards,

Yitagesu Habtu Aweke, Ph.D

Academic Editor

PLOS ONE

Journal Requirements:

**Additional Editor Comments:**

 In addition to the reviewer's minor issues, you may consider the following comments: 

In your abstract section, "Primary healthcare (PHC) facilities have become essential in promoting adolescent healthcare, yet they face resource limitations that hinder their effectiveness. Ensuring the efficient use of available resources has therefore become pertinent.” Yes, it is important, but what was your research questions?  Describing the research question (gaps) that led you to assess the technical efficiency of primary health care facilities benefits the clarity before you state your objective (i.e., This study assessed.....)Numbering headings, and sub-heading adds minimal benefits to the clarity of your content in some sections. So, it’s important to think over removing numbering of headings and subheadings  or  collapsing some third-level headings into second level, and succinctly describing them benefit the clarity and organization of the manuscript. e.g. collapse "2.3.1 "  in your method section into  2.3, and do the same thing in your result section, eg. 3.2.1.... Finally, avoid the sub-sectioning, and continue as “Introduction”, “methods”, “results”, Discussion”, “conclusion”...Break your  "3.0 Results and discussion" into separate levels  as " Results", " Discussion"Your discussion seems to lack depth. Please first, summarize what you learned from the results, and discuss and compare your findings with other studies in Ghana. There are studies within Ghana, and other countries with nearly similar contexts,  possible bias, limitations, policy implications etc

Reviewers' comments:

Reviewer's Responses to Questions

**Comments to the Author**

1. If the authors have adequately addressed your comments raised in a previous round of review and you feel that this manuscript is now acceptable for publication, you may indicate that here to bypass the “Comments to the Author” section, enter your conflict of interest statement in the “Confidential to Editor” section, and submit your "Accept" recommendation.

Reviewer #3: (No Response)

2. Is the manuscript technically sound, and do the data support the conclusions?

Reviewer #3: Yes

3. Has the statistical analysis been performed appropriately and rigorously? 

Reviewer #3: Yes

4. Have the authors made all data underlying the findings in their manuscript fully available?

Reviewer #3: Yes

5. Is the manuscript presented in an intelligible fashion and written in standard English?

Reviewer #3: No

6. Review Comments to the Author

Reviewer #3: Dear authors,

Thank you for the opportunity to review this new submission, and for your detailed responses to my comments and those of the other reviewers. As I wrote earlier, I enjoyed reading this article for your study on the technical efficiency of adolescent health services in Ghana.

In the introduction, please carefully review the formatting for citations, since several are outside of the sentences to which they belong.

In the materials and methods section, please define DEA (line 126) and add some context to what "flexibility" means for the translog function (line 151).

For the results, in Table 2, please clarify what "Insurance" as a characteristic means. Also, I want to reiterate that, for lines 262-273, for many of the facility characteristics, the differences in mean efficiency scores is very small and the confidence intervals overlap. For example, the mean score for rural facilities is 0.630, but that score is fits in the confidence interval for urban facilities. Therefore, I am unsure if you can say that they differences are statistically significant, as you do in lines 272-273. This is even more pronounced--that the mean scores are very narrowly different--for characteristics like facility laboratory or gender of facility head.

Finally, for the discussion, I saw that you revised the writing around fieldwork observations. I suggest adding some language to couch the statements as from your or the data collection team's perspectives. For example, "We heard from adolescent patients that they would prefer to self-medicate..." While I saw that you addressed my comment about sample size, I recommend that you discuss this as a limitation to the study. I am not familiar with the SFA, but I wonder whether ~50 firms in the analytic sample is sufficient, especially with the 20 covariates in the final models. I looked around and found some studies that simulate SFA models under different sample sizes, like Cheng et al. (2024): https://www.sciencedirect.com/science/article/pii/S0304407622000677.

Throughout the paper, there are still grammatical errors and formatting issues that need to be addressed, including extra spaces, missing periods, incomplete sentences, etc.

Thank you!

7. PLOS authors have the option to publish the peer review history of their article (what does this mean? ). If published, this will include your full peer review and any attached files.

**Do you want your identity to be public for this peer review?** For information about this choice, including consent withdrawal, please see our Privacy Policy .

Reviewer #3: No

---

## [Author Response · Author response to Decision Letter 1]

3 Mar 2025

This has been attached as a separate document.

---

## [Editor Report · Decision Letter 2]

4 Mar 2025

Technical Efficiency of Primary Health Facilities in Providing Adolescent Mental, Sexual and Reproductive Health Services in Ghana: a case study of selected districts in the Greater Accra Region

PONE-D-24-09524R2

Dear Dr. Ama Pokuaa Fenny,

We’re pleased to inform you that your manuscript has been judged scientifically suitable for publication and will be formally accepted for publication once it meets all outstanding technical requirements.

Kind regards,

Yitagesu Habtu Aweke, Ph.D

Academic Editor

PLOS ONE

Additional Editor Comments (optional):

Although they don't have much impact on the scientific content, and readability effect, it is good if you are able to:

  avoid numbering of  "headings" and "subheading"  because you have a maximum of two headings

avoid "Materials" from the "Materials and Methods"

remove "Acknowledgements"  if you don't have anybody to acknowledge,  instead of writing "Not applicable".

---

## [Editor Report · Acceptance letter]

PONE-D-24-09524R2

PLOS ONE

Dear Dr. Fenny,

I'm pleased to inform you that your manuscript has been deemed suitable for publication in PLOS ONE. Congratulations! Your manuscript is now being handed over to our production team.

Kind regards,

on behalf of

PhD Candidate Yitagesu Habtu Aweke

Academic Editor

PLOS ONE